# Lack of Benefit of Extending Temozolomide Treatment in Patients with High Vascular Glioblastoma with Methylated *MGMT*

**DOI:** 10.3390/cancers13215420

**Published:** 2021-10-29

**Authors:** María del Mar Álvarez-Torres, Elies Fuster-García, Carmen Balaña, Josep Puig, Juan M. García-Gómez

**Affiliations:** 1Biomedical Data Science Laboratory, Instituto Universitario de Tecnologías de la Información y Comunicaciones, Universitat Politècnica de València, 46022 Valencia, Spain; elies.fuster@gliohab.eu (E.F.-G.); juanmig@ibime.upv.es (J.M.G.-G.); 2Department of Diagnostic Physics, Oslo University Hospital, 0450 Oslo, Norway; 3Institut Catala d’Oncologia (ICO), Applied Research Group in Oncology (B-ARGO Group), Institut Investigació Germans Trias i Pujol (IGTP), 08916 Badalona, Spain; cbalana@iconcologia.net; 4Institut de Diagnostic per la Image (IDI), Hospital Dr. Josep Trueta, 17007 Girona, Spain; jpuigmd@gmail.com

**Keywords:** glioblastoma, *MGMT* methylation, tumor vascularity, chemotherapy, adjuvant temozolomide, temozolomide cycles, MRI perfusion, rCBV, survival, personalized medicine

## Abstract

**Simple Summary:**

Despite the complete treatment with surgery, chemotherapy and radiotherapy, patients with glioblastoma have a devasting prognosis. Although the role of extending temozolomide treatment has been explored, the results are inconclusive. Recent evidence suggested that tumor vascularity may be a modulating factor in combination with methylation of O6-methylguanine-DNA methyltransferase (*MGMT*) promotor gene on the effect of temozolomide-based therapies, opening new possibilities for personalized treatments. Before proposing a prospective interventional clinical study, it is necessary to confirm the beneficial effect of the combined effect of *MGMT* methylation and moderate tumor vascularity, as well as the lack of benefit of temozolomide in patients with a highly vascular tumor.

**Abstract:**

In this study, we evaluated the benefit on survival of the combination of methylation of O6-methylguanine-DNA methyltransferase (*MGMT*) promotor gene and moderate vascularity in glioblastoma using a retrospective dataset of 123 patients from a multicenter cohort. MRI processing and calculation of relative cerebral blood volume (rCBV), used to define moderate- and high-vascular groups, were performed with the automatic ONCOhabitats method. We assessed the previously proposed rCBV threshold (10.7) and the new calculated ones (9.1 and 9.8) to analyze the association with survival for different populations according to vascularity and *MGMT* methylation status. We found that patients included in the moderate-vascular group had longer survival when *MGMT* is methylated (significant median survival difference of 174 days, *p* = 0.0129*). However, we did not find significant differences depending on the *MGMT* methylation status for the high-vascular group (*p* = 0.9119). In addition, we investigated the combined correlation of *MGMT* methylation status and rCBV with the prognostic effect of the number of temozolomide cycles, and only significant results were found for the moderate-vascular group. In conclusion, there is a lack of benefit of extending temozolomide treatment for patients with high vascular glioblastomas, even presenting *MGMT* methylation. Preliminary results suggest that patients with moderate vascularity and methylated *MGMT glioblastomas* would benefit more from prolonged adjuvant chemotherapy.

## 1. Introduction

Glioblastoma patients remain a devastating prognosis of 12–15 months from diagnosis [1,2] despite an intrusive treatment including tumor resection, radiotherapy, and concomitant and maintenance chemotherapy with temozolomide [3]. This standard treatment, proposed by Stupp in 2005 [3], was demonstrated to be the most effective in terms of overall survival but, due to strong interpatient heterogeneity, it is not equally efficient for all patients [4].

Several studies have evaluated the efficacy of this treatment depending on several conditions as extend of tumor resection [5,6,7,8,9,10,11], age [12], the methylation of the O6-methylguanine-DNA methyltransferase (*MGMT*) promoter gene [13], dose of temozolomide [14,15,16,17,18,19,20], the addition of new agents [21,22,23], or the device tumor treating fields [24,25,26,27,28,29,30], that in fact is the only modification that has proven to increase survival.

The optimal number of cycles of temozolomide in the maintenance phase has also been a matter of debate [31,32]. This is due to the heterogeneity of uses or interpretation of the term ‘maintenance’ or ‘adjuvant therapy’ in a disease such as glioblastoma where surgery seldom achieves a complete resection. The number of cycles administered is clearly variable in the clinical setting or even in the different trials [33]. The only prospective trial assessing the role of extending temozolomide further than six cycles is a randomized phase II trial that did not demonstrate differences in progression-free survival or overall survival [14]. The European Association of Neuro-oncology guidelines recommend six cycles of maintenance therapy [34].

The same treatment for all patients with glioblastoma has been demonstrated ineffective. The availability of robust markers to characterize interpatient heterogeneity, and, therefore, to discriminate different subgroups could lead to a more personalized medicine approach. In this line, imaging markers derived from MRI and combined with the capabilities of artificial intelligence can provide individually specific variations of the Stupp treatment. This would allow better prognosis and facilitate the clinical decision-making for patient treatment in a non-invasive way and without additional cost [35,36,37,38,39].

Currently, glioma classification, decision making, and management of glioblastoma are still based on molecular biomarkers [1,40,41,42,43]. One of the most relevant biomarkers, related with the Stupp treatment efficacy, is *MGMT* methylation status [44], present in approximately 50% of glioblastomas [45]. *MGMT* removes alkyl groups from guanine in the DNA, potentially counteracting the therapeutic efficacy of alkylating chemotherapeutics, such as temozolomide, in tumor cells [43,46]. Methylation of the promoter region of *MGMT* might lead to transcriptional repression and a decreased *MGMT* protein expression [43,46]. It is associated with an improved response to temozolomide chemotherapy and longer overall survival of GBM patients [1,40,41,42,43].

A recent study with a multicenter cohort of 96 glioblastoma patients [47] concluded that *MGMT* methylation may benefit overall survival only in patients with moderately vascularized glioblastomas, defined by MRI perfusion-based marker, such as relative cerebral blood volume (rCBV). This study opened the possibility of investigating vascularity as a determinant factor, in combination with methylation status, on the benefit of temozolomide cycles.

In this study, we aimed to evaluate the combined effect of *MGMT* methylation and tumor vascularity on patient survival, assessing the performance of the proposed rCBV threshold for patient stratification. We also assessed the implications of the association between *MGMT* methylation and vascularity on the benefit of extending temozolomide treatment in different groups of glioblastoma patients.

## 2. Materials and Methods

### 2.1. Patient Information

For this study, 123 glioblastoma patients were included from the GLIOCAT database [48], which includes patients from the following six centers from Cataluña, Spain: (I) Instituto Catalán de Oncología (ICO) de Badalona (Barcelona), (II) Hospital del Mar (Barcelona), (III) Hospital Clínic (Barcelona), (IV) ICO Hospitalet (Barcelona), (V) ICO Girona (Girona), and (VI) Hospital Sant Pau (Barcelona). A Material Transfer Agreement was approved by all the participating centers and an acceptance report was issued by the Ethical Committee of each center.

The inclusion criteria were: (a) adult patients (age >18 years) with histopathological confirmation of glioblastoma; diagnosed between June 2007 and May 2015, (b) with access to the preoperative MRI studies, including: pre- and post-gadolinium T1-weighted, T2-weighted, Fluid-Attenuated Inversion Recovery (FLAIR), and Dynamic Susceptibility Contrast (DSC) T2*-weighted perfusion sequences; (c) with *MGMT* methylation status information, (d) with a minimum survival of 30 days and, (e) with tumor resection.

Patients still alive at readout were considered censored observations. The date of censorship was the last date of contact with the patient or, if not available, the date of the last MRI exam.

The patient cohort included in this study is totally independent from that which was analyzed in the previous study [47].

### 2.2. Magnetic Resonance Imaging

Standard-of-care MR examinations were obtained for each patient before surgery, including pre- and post-gadolinium-based contrast agent enhanced T1-weighted MRI, as well as T2-weighted, FLAIR T2- weighted, and DSC T2* perfusion MRI.

### 2.3. MRI Processing and Vascular Marker Calculation

To process the MRIs and calculate the imaging vascular markers, we used the Hemodynamic Tissue Signature (HTS) method [49,50], freely accessible at the ONCOhabitats platform at www.oncohabitats.upv.es, accessed on 4th June 2020. The HTS is an automated unsupervised method developed to describe the heterogeneity of the enhancing tumor and edema tissues at morphological and vascular levels, and to calculate robust biomarkers with prognostic and patient stratification capabilities. This method includes the following four phases (Figure 1):MRI Pre-processing. This phase includes voxel isotropic resampling of all MR images, correction of the magnetic field in homogeneities and noise, rigid intra-patient MRI registration, and skull-stripping.Glioblastoma tissue segmentation. It is performed using an unsupervised segmentation method, which implements a state-of-the-art deep-learning 3D convolutional neural network (CNN), which takes as input the T1c, T2, and Flair MRIs. This method is based on Directional Class Adaptive Spatially Varying Finite Mixture Model, or DCA-SVFMM, which consists of a clustering algorithm that combines Gaussian mixture modeling with continuous Markov Random Fields to take advantage of the self-similarity and local redundancy of the images.DSC perfusion quantification. In this phase, biomarkers such as the relative cerebral blood volume (rCBV) maps, as well as relative cerebral blood flow (rCBF) or Mean Transit Time (MTT), are calculated for each patient. T1-weighted leakage effects are automatically corrected using the Boxerman method [51], while gamma-variate curve fitting is employed to correct for T2 extravasation phase. rCBV maps are calculated by numerical integration of the area under the gamma-variate curve. The Arterial Input Function (AIF) is automatically quantified with a divide and conquer algorithm.Hemodynamic Tissue Signature and Vascular Habitats. The HTS provides an automated unsupervised method to describe the heterogeneity of the enhancing tumor and edema tissues, in terms of the angiogenic process located at these regions. We consider four sub-compartments for the glioblastoma, two within the active tumor: High Angiogenic Tumor habitat (HAT) and Low Angiogenic Tumor habitat (LAT); and two within the edema: Infiltrated Peripheral Edema habitat (IPE) and Vasogenic Peripheral Edema habitat (VPE). These four habitats are obtained by the unsupervised analysis of perfusion patterns, which is carried out through the Directional Class Adaptative Spatially Varying Finite Mixture Model (DCA-SVFMM) algorithm. Such algorithm is an extension of the classic FMM specially focused on image data, which incorporates a continuous Markov Random Field on the spatial coefficients of the model to capture the self-similarity and local redundancy of the images. The clustering consists of two stages: (a) a two-class clustering of the whole enhancing tumor and edema ROIs and (b) a two-class clustering performed using only the rCBV and rCBF data within the ROIs obtained in stage a to detect the different vascular behaviors expressed by the glioma.

A more detailed description of the methodology is included in [49,50]. In addition, the HTS method and the vascular biomarkers were validated in an international multicenter study and results were published in [52].

To validate the combined effect of *MGMT* methylation and vascularity, we used the maximum relative cerebral blood volume (rCBVmax) calculated in the HAT habitat, since it is shown to be the most relevant prognostic marker calculated with the HTS method [50,52,53] and it was used in the previous study to define the vascular groups [47].

### 2.4. Moderate- and High-Vascular Groups

The entire cohort was divided in two groups according to the tumor vascularity: the moderate-vascular group and the high-vascular group. To determine these groups, we carried out the analysis independently using three different thresholds (th) of the HAT rCBVmax:(I).The threshold proposed in the literature (49) (th = 10.7). It was calculated as the median rCBVmax of 96 patients included in an international multicenter study.(II).The median rCBVmax of the current study cohort (th = 9.1). Calculated from the 123 patients included in the present study.(III).The combined threshold of both cohorts (th = 9.8). It is calculated considering the 219 patients from two independent multicenter studies.

The purpose to evaluate these three different thresholds is to validate the previous results and the threshold proposed in the literature [47] with an independent multicenter cohort; but also to analyze the stratification capability of the HAT rCBVmax when using the specific threshold calculated from the current study cohort. Finally, proposing a combined threshold calculated from both independent cohorts with 219 patients will allow most reproducible results.

### 2.5. DNA Extraction and Assessment of MGMT Methylation

DNA was extracted from two 15-µm sections of FFPE tissue using the QIAamp DNA Mini Kit (QIAGEN GmbH, Hilden, Germany), following the manufacturer’s protocol. In cases with less than 50% of tumor cells, the tumor tissue was macro-dissected manually. Then 500 ng of extracted DNA was subjected to bisulfite treatment using the EZ DNA Methylation-Gold Kit (Zymo Research Corporation, Irvine, CA, USA). DNA methylation patterns in the CpG island of the *MGMT* gene were determined by methylation-specific PCR (MSP) using primers specific for either methylated or modified non-methylated DNA, as previously described [54].

### 2.6. Statistical Analyses

#### 2.6.1. Dataset Description: Differences between Methylated and Unmethylated *MGMT* Groups

We described the main demographic, clinical, and molecular variables for the entire cohort and for methylated and unmethylated *MGMT* populations. The analyzed variables for each population were: gender, age at diagnosis, survival times, extent of tumor resection, completeness of concomitant chemotherapy, number of adjuvant temozolomide cycles, *IDH1* mutation status, and rCBV_max_ at HAT habitat. Possible differences in the distributions of these variables for the populations with methylated and unmethylated *MGMT* were assessed using Mann–Whitney U test (for ordinal or continuous variables) or Fisher exact test (for nominal variables) in MATLAB R2017b (MathWorks, Natick, MA, USA). The significance level used in all the statistical analyses was 0.05.

#### 2.6.2. Association between *MGMT* Methylation, Tumor Vascularity and Patient Survival

To validate the previous results published in [47], which showed a significant correlation between *MGMT* methylation status with overall survival only for those patients with moderate vascularized tumors, we carried out the Uniparametric Cox proportional hazard regression. These analyses were carried out for the entire cohort, and independently for the methylated and unmethylated *MGMT* groups and using the three studied thresholds. The proportional hazard ratios (HRs) with a 95% confidence interval (CI), as well as the associated *p*-values are reported.

#### 2.6.3. Survival Differences between Groups According to Tumor Vascularity and *MGMT* Methylation Status

Kaplan Meier test was carried out to evaluate the different effect on survival of *MGMT* methylation status, depending on tumor vascularity and, the Log rank was used to determine any statistical differences between the estimated survival functions of the different *MGMT* methylation populations, both at moderate- and high-vascular groups. The number of patients included in each group, the median OS rates of each group, the differential OS, and the *p*-values are reported.

The following results were carried out using the (III) combined threshold (th = 9.8), since authors consider it as the most robust threshold because its calculation was derived from data of 214 patients from two different multicenter datasets and could generate more repeatable results.

#### 2.6.4. Benefit of Adjuvant Temozolomide Cycles in Different Groups of Glioblastoma Patients

To analyze the combined effect of *MGMT* methylation and the number of adjuvant temozolomide cycles on survival, a Multiparametric Cox regression analysis was carried out including *MGMT* methylation status and number of temozolomide cycles for the entire cohort, and independently for the moderate- and high-vascular groups. The number of temozolomide cycles was a continuous variable with a minimum of 0 to a maximum of 12 cycles.

In addition, to study differences in patient survival associated with the number of administered temozolomide cycles, a boxplot was carried out for each group (defined by *MGMT* methylation status and tumor vascularity).

## 3. Results

### 3.1. Study Cohort

This study includes data from 123 patients with primary glioblastoma (Table 1).

Any variable was found as statistically different between methylated and unmethylated MGMT groups (*p* < 0.05), suggesting that any of these variables affect the results of the rest of survival and stratification analyses.

### 3.2. Lack of Benefit of Temozolomide for MGMT Methylated Patients with High Vascular Tumors

#### 3.2.1. Uniparametric Cox Regression Analysis

Table 2 includes the results of the Uniparametric cox regression analyses for the entire cohort and for the moderate- and high-vascular groups, generated with different proposed cut off thresholds: (I) the threshold proposed in the preliminary study [47], (II) the median HAT rCBVmax of the current study cohort, and (III) the threshold calculated with the combination of both cohorts (n = 214 patients).

The Uniparametric Cox results show a significant association between the MGMT methylation status and patient overall survival (OS) for the entire cohort of 123 patients. However, when this association is analyzed individually for the moderate- and high-vascular groups, we only found significant results for the group of patients with moderate rCBV, regardless of the threshold used. By contrast, we did not find a significant association for the group with high rCBV. These results are repeated for all the vascular groups generated with the three analyzed thresholds, although they are more patent when using the specific threshold of the study cohort, yielding higher HR and lower *p*-value.

#### 3.2.2. Kaplan Meier and Log Rank Test

Kaplan Meier results for the entire cohort and for the moderate- and high-vascular groups are included in Table 3.

The Kaplan Meier results showed significant differences in survival for the entire cohort (*p* < 0.05) depending on the MGMT methylation status. However, these differences in survival time were more significant (lower *p*-value) and more patent (higher difference in survival days) for the moderate-vascular group. For this group, we found significant differences (*p* = 0.0129) in median survival between the populations with methylated MGMT and with unmethylated MGMT (641 vs. 467 days, respectively), with a difference in OS of 174 days. By contrast, we did not find any difference in survival for the high-vascular group, independently of their MGMT methylation status.

This differential effect of MGMT methylation depending on tumor vascularity is also illustrated in Figure 2, which shows the Kaplan Meier survival curves for each vascular group and for each MGMT population. The Kaplan Meier curves using the other two proposed thresholds are included in Appendix A of the Appendix A.

The Kaplan Meier curves reaffirm the results that the influence of MGMT methylation on survival time is only for the moderate vascular group, since only for this group are the survival functions significantly different.

### 3.3. Benefit of Adjuvant Temozolomide Cycles in Different Groups of Glioblastoma Patients

#### Multiparametric Cox Regression Analysis

Multiparametric Cox results for the entire cohort, and independently for the moderate- and high-vascular groups, including hazard ratios, Cis, and *p*-values are shown in Table 4.

A significant correlation between the number of temozolomide cycles and patient survival was found for the entire cohort, and for the moderate- and high-vascular groups. Nonetheless, only for the moderate vascular group a significant correlation for both variables (MGMT methylation status and number of temozolomide cycles) was found. These results suggest that the combined effect of these two clinical variables is more relevant for survival time for those patients with moderate tumor vascularity.

Additionally, Figure 3 shows a boxplot per each following group, with survival times depending on the number of adjuvant temozolomide cycles administered:(a)Moderate vascularity + methylated MGMT(b)Moderate vascularity + unmethylated MGMT(c)High vascularity + methylated MGMT(d)High vascularity + unmethylated MGMT

We can see that for the unmethylated MGMT populations (in green and in red), median survival rates do not overcome 700 days in any case, independently from adjuvant temozolomide cycles.

By contrast, different tendencies could be appreciated for the methylated MGMT populations. In the case of patients with moderate vascularity (in blue), median survival rates seem to increase with higher number of temozolomide cycles, being the highest median OS for the group with more than six temozolomide cycles.

However, the tendency seems different for the high vascular group (in yellow). Although patients that completed the standard six-cycle treatment, presented a higher survival rate, to administer more than six cycles do not seem to provide a beneficial effect, even an adverse one.

## 4. Discussion

With the present study, we aimed to evaluate the lack of benefit of temozolomide for *MGMT* methylated patients with high vascular glioblastomas, since previous results published in [47] concluded that the combined effect of *MGMT* methylation and moderate vascularity of the tumor causes a benefit in glioblastoma patient overall survival. For this purpose, we have analyzed data from an independent and larger cohort than in [47]. In addition, the previously proposed threshold has been validated and we propose an upload to be more generalizable in future studies, since it has been calculated with data from 214 patients. Finally, we aimed to investigate the potential benefit of increasing the number of adjuvant temozolomide cycles in different groups of glioblastoma patients according to their *MGMT* methylation status and tumor vascularity.

To achieve our main purposes, we used an independent and major multicenter cohort of 123 glioblastoma patients. Our results validate the hypothesis proposed in [48], since we have found significant associations (*p* < 0.05) between *MGMT* methylation status and patient survival only for the moderate vascular group of patients, but not for the high vascular group (*p* > 0.05). This is, prognosis of patients with a moderate vascular tumor will be affected by *MGMT* methylation status, while survival times for the high vascular group do not differ, independently of the *MGMT* methylation status. This evidence is also shown when analyzing the Kaplan Meier results: for the moderate-vascular group there is a significant difference (*p* < 0.005) of 174 days in median survival depending on presenting methylated or unmethylated *MGMT*, while for the high-vascular group there are not significant differences in survival. That is, there is a lack of benefit of temozolomide for *MGMT* methylated patients with high vascular glioblastomas.

Some clinical studies have been developed with the purpose of analyzing the effect of increasing the number of adjuvant temozolomide cycles [14,15,16,17,18,19,20], six cycles being considered as the standard [3]. One meta-analysis [31] and a retrospective large cohort analysis [32] found no benefits on OS but a possible improvement in progression free survival. The only randomized phase II trial did not show any benefit in those parameters for the fact of continuing temozolomide for further than six cycles. Anyway, this was only a phase II trial with a small number of patients, and it may be that a particular subgroup of patients gets benefits from continuing temozolomide treatment, as our preliminary results suggest.

Considering previous results, which opened the possibility to investigate the different effect of temozolomide in particular subgroups, we explored the benefit of increasing the number of temozolomide cycles depending on their specific *MGMT* status and vascular profile. We investigated the correlation between the number of temozolomide cycles and *MGMT* status for the high- and moderate-vascular groups and we found that only for the moderate-vascular group, both variables were significantly associated with patient survival. We hypothesize that tumors with a lower vascularization could be potentially less aggressive, with a lower prevalence of molecular aberrations that may confer resistance to alkylating agents in the presence of a methylated *MGMT*. Additionally, a high tumor vascularity could be related with a faster progression, hindering the damaging effect of temozolomide on tumor cells. In this sense, advanced MRI-based methodologies can complement molecular analysis to help in glioblastoma characterization and therapy selection [54,55,56,57].

Furthermore, we analyzed, in an observational way, the survival patterns of each group (defined by *MGMT* status and vascularity) and with different number of administered temozolomide cycles (<6, 6 or >6). We found specific survival tendencies for each group of patients when administering different number of temozolomide cycles. The group of patients with methylated *MGMT* and moderate vascularity was observed as the only one that benefits from more than six temozolomide cycles. Additionally, we want to highlight the low number of patients who received more than six temozolomide cycles, being less than 10% of the entire cohort. This fact is probably due to the retrospective nature of our cohort when more than six cycles were administered in some centers before the evidence generated in subsequent studies [14,47].

These are preliminary results but considering the interest in deciding more individual treatments for glioblastoma patients, future prospective studies could be relevant to analyze the beneficial effect of providing more than six cycles of temozolomide for selected groups of patients. Knowing the marked interpatient heterogeneity, a more personalized approach to treat glioblastoma patients appears to be a potential solution to overcome the heterogeneity and prolonged overall survivals.

The main limitation of this study is the lack of a randomized strategy to provide more than six cycles of adjuvant temozolomide to patients. This is due to the observational and retrospective nature of the study. Assuming that association does not imply causation, our results of analyzing the prognostic effect of temozolomide cycles should be interpreted with caution. In addition, despite the sample size is large enough comparing with previous studies; the comparison of the effect of different number of TMZ cycles could be affected by a small size in some groups of patients, since 12 different groups have been analyzed. However, differences in survival tendencies among groups seem to exist, and future prospective studies could validate these results. These limitations are only referred to the second objective of the study.

## 5. Conclusions

In conclusion, our results demonstrate the lack of benefit of extending temozolamide treatment in those patients with high vascular glioblastoma, even presenting *MGMT* methylation. In addition, we have validated the previously proposed threshold (th = 10.7) as useful to stratify patients in terms of vascularity and with significant differences in survival, and we proposed an upload threshold, calculated with both cohorts, (th = 9.8) to be more generalizable in future studies. Finally, we found preliminary results related with the potential benefit of increasing the number of adjuvant temozolomide cycles only for a particular group of patients with *MGMT* methylation and moderate vascularity, which represents almost a 40% of the study entire cohort. Authors consider clinically relevant a future prospective study analyzing the beneficial effect of providing more than six temozolomide cycles in the group of patients with moderate vascularity and methylated *MGMT*. Positive results could lead us to a more personalized decision making in glioblastoma treatment, in particular during the chemotherapy stage, allowing prolonged survival times in patients with methylated MGMT and moderate vascular tumors and avoiding toxicity in patients with high vascular tumors.

## Figures and Tables

**Figure 1 cancers-13-05420-f001:**
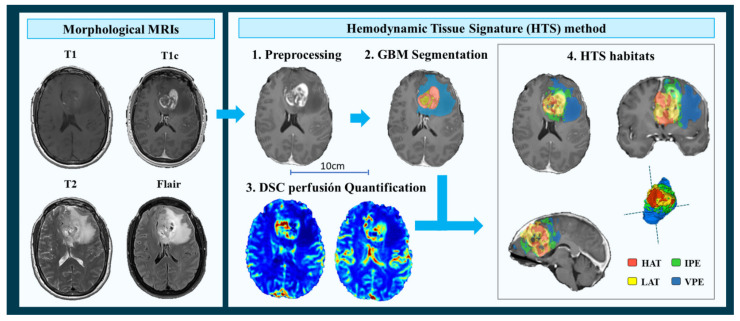
Hemodynamic Tissue Signature (HTS) method, including the four phases: 1. Preprocessing of morphological MRIs (T1, T1c, T2, and Flair); 2. Glioblastoma tissue segmentation; 3. DSC perfusion quantification; and 4. HTS vascular habitats. HAT: High Angiogenic Tumor, LAT: Low Angiogenic Tumor, IPE: Infiltrated Peripheral Edema, and VPE: Vasogenic Peripheral Edema.

**Figure 2 cancers-13-05420-f002:**
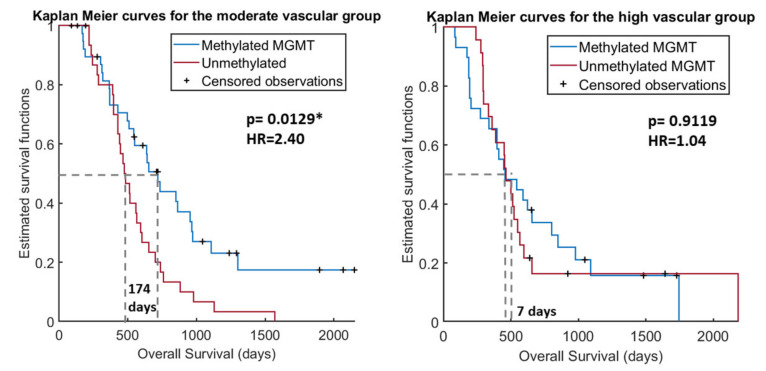
Kaplan Meier curves for the moderate-vascular group (**A**) and for the high-vascular group (**B**) depending on the MGMT methylation status. * Significant *p*-values: < 0.05.

**Figure 3 cancers-13-05420-f003:**
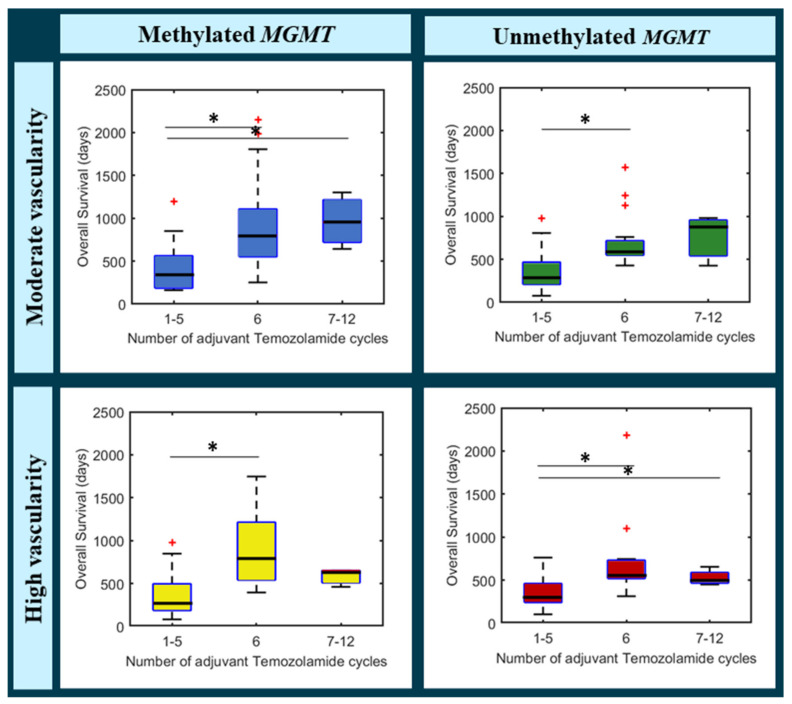
Boxplots analyzing the differences in overall survival according to the administered number of adjuvant Temozolomide cycles (1–5, 6, and 7–12) for different populations: patients with moderate vascularity and methylated MGMT (top left), patients with moderate vascularity and unmethylated MGMT (top right), patients with high vascularity and methylated MGMT (bottom left), and patients with high vascularity and unmethylated MGMT. * Significantly differences between groups (* in black). Data points beyond the whiskers are shown with + (in red).

**Table 1 cancers-13-05420-t001:** Demographic, clinical, and biological characteristics of the entire cohort, and the groups with methylated and unmethylated MGMT. *p*-values derived from Mann–Whitney (MW) test or Fisher exact (FE) test analyzing differences between methylated and unmethylated MGMT.

Variables	Entire Cohort	Methylated *MGMT* Population	Unmethylated *MGMT* Population	*p*-Values (MW/FE)
**Number of patients**	123	67	56	-
**Gender**				
-% females	41.5	43.3	39.2	0.7150
**Age at diagnosis (years)**				0.8973
-Mean	60	62	58	
-Range	(32,80)	(33,80)	(32,77)	
**Overall Survival (months)**				0.1214
-Mean	20.2	22.4	17.6	
-Median	17.1	19.3	15.5	
-Range	(2.7,72.8)	(2.7,71.6)	(2.7,72.8)	
**Extent of Resection (#patients)**				0.4524
-Complete	45	27	18	
-Partial	78	40	38	
**Concomitant chemotherapy (#patients)**				0.3788
-Complete	110	58	52	
-Incomplete	13	9	4	
**Adjuvant chemotherapy (number of cycles)**				0.4435
-Mean	4	5	4	
-Median	5	5	4	
-Range	(0,12)	(0,12)	(0,12)	
**IDH1 mutation status**				1.0000
-Mutated	2	1	1	
-Wild type	93	51	42	
-Unknown	28	15	13	
**HAT rCBV_max_**				0.4150
-Mean	9.77	9.49	10.10	
-Median	9.10	9.53	8.87	
-Range	(3.39, 21.80)	(3.39, 16.93)	(3.49, 21.8)	

**Table 2 cancers-13-05420-t002:** Uniparametric Cox regression results for the entire cohort, and for the moderate- and high-vascular groups, using different proposed cut off thresholds: (I) the threshold proposed in the preliminary study [ref], (II) the median rCBVmax of the present study cohort, and (III) the combined threshold calculated with data of both populations (n = 214 patients).

Association *MGMT* Methylation–Overall Survival		Number of Patients	HR [95% CI]	*p*-Value
**Analyzed thresholds**	Entire cohort	123	1.58 [1.06, 2.35]	0.0247 *
(I) Th. proposed in [47] = 10.7	Moderate rCBV	80	**1.70** [1.04, 2.79]	0.0353 *
	High rCBV	43	1.36 [0.69, 2.67]	0.3734
(II) Th. study cohort = 9.1	Moderate rCBV	61	**2.40** [1.34, 4.31]	0.0032 *
	High rCBV	62	1.04 [0.60, 1.80]	0.9008
(III) Th. combined = 9.8	Moderate rCBV	71	**2.01** [1.19, 3.41]	0.0095 *
	High rCBV	53	1.09 [0.59, 2.00]	0.7894

***** Significant *p*-values: < 0.05. Th.: threshold; HR: Hazard Ratio; CI: Confidence Interval.

**Table 3 cancers-13-05420-t003:** Kaplan Meier results for the entire cohort, and for the moderate- and high-vascular groups, generated by the combined cut off threshold (9.8), when comparing the populations with methylated and unmethylated MGMT.

Survival Rates According to *MGMT* Methylation and Tumor Vascularity	Number of Patients			KM Results			
Kaplan-Meier Analysis	Total	Meth. *MGMT*	Unmeth. *MGMT*	Median OS Meth. *MGMT*	Median os Unmeth. *Mgmt*	|ΔOS|	*p*-Value
Entire cohort	123	67	56	578	462	114	0.0220 *
Moderate rCBV	71	46	34	641	467	174	0.0129 *
High rCBV	53	21	22	454	461	7	0.9119

* Significant *p*-values: < 0.05. Meth. MGMT: methylated MGMT; Unmeth. MGMT: unmethylated MGMT; OS: overall survival.

**Table 4 cancers-13-05420-t004:** Multiparametric Cox regression results for the entire cohort, and for the moderate- and high-vascular groups, analyzing the combined correlation between the MGMT methylation status and the number of adjuvant Temozolomide-cycles with the overall survival.

Covariables	HR [95% CI] MGMT	*p*-Value
**Entire cohort**		
*MGMT* status	1.53 [0.96, 2.43]	0.0727
TMZ cycles	0.78 [0.70, 0.85]	<0.0001 *
Moderate rCBV		
*MGMT* status	1.75 [1.08, 4.20]	0.0416 *
TMZ cycles	0.78 [0.66, 0.90]	<0.0001 *
High rCBV		
*MGMT*	1.03 [0.56, 1.92]	0.9121
TMZ cycles	0.77 [0.68, 0.87]	<0.0001 *

***** Significant *p*-values: < 0.05.

## Data Availability

The data presented in this study are available in this article (and Appendix A).

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
