# Peer review of "Lack of Benefit of Extending Temozolomide Treatment in Patients with High Vascular Glioblastoma with Methylated MGMT"

_cancers, 2021, doi:10.3390/cancers13215420_

Round 1

Reviewer 1 Report

The authors applied habitat imaging concept to study the response to temozolomide for GBM patient with MGMT. The study is well presented. However there are few comments I'd like to share and improve.

First, the proposed vasculature habitat is sensitive the the cutoff value. Here, the authors did a good work to validate previously proposed cutoff. After that, it is interesting to pool the previous and current samples together to see if the cutoff value can be further optimized. 

Second, there are several technical improvements can be considered in the future work. In our previous work in this same field, we define the habitat using multiparametric MRIs (PMID: 28168370). Moreover, beyond using cutoff values to define habitats, we also proposed to leverage unsupervised clustering to define intratumoral heterogeneity (PMID: 29714680). Lately, we defined new imaging subtypes and showed that they are prognostic in GBM (https://www.nature.com/articles/s42256-021-00377-0).

Reviewer 2 Report

In the article presented, the authors discovered a good opportunity for potential improvement in subgroups of glioblastoma patients by simply increasing the cycles of standard treatment for those who have moderate vascularity. Through a prospective study, these findings would be a great contribution to the area of precision medicine, if this does indeed prolong survival. However, there are some other comments that need clarification, and a few grammatical corrections.

Comment 1:

It is unclear whether the finding of the impact of methylated or unmethylated status and vascularity on survival has been confirmed in a different cohort in the current study (compared to previous cohort from [47] - are the authors presenting the same data?  It is a little unclear – please clarify. If it is the same, then the discussion regarding this should be reduced, and have more of a focus on the new information.

Comment 2:

Figure 3: Boxplot: When the authors identify that the high-vascular tumour group (methylated group) do not benefit from temozolomide – could this be due to the sample size?  The boxplot still shows a much higher OS with 6 cycles for the methylated and high vascularity group compared to the 0-5 cycles – and I cannot see the CIs as it is not clearly presented – not sure if this is an accident but it should be fixed.  

Comment 3:

In general the authors conclude that there is a lack of benefit for treating high vascular glioblastoma patients with TMZ – can they clarify, do they mean 6 cycles or more, or are they suggestion that TMZ should potentially not be used for this subgroup at all?

Grammar:

Introduction:

Line 59: ‘A same treatment’ should be changed to ‘the same‘

Line 65: ‘way and not-extra cost [35-39]’ should be changed to ‘without additional cost‘

Line 69: ‘repair DNA damaged’ should be changed to ‘repair damaged DNA’

In section 2.3.4, the authors jump from writing out the number is full and then not. ‘4’, and ‘two’ – please be consistent.

Results:

Line 217-220: delete the repetition because you repeat the description twice, both in the paragraph and also in the table title (you also do this in section 3.2.1 – no need to write it twice:

Delete:  ‘sum- 217

marizes the main demographic, clinical and biological characteristics of the entire cohort 218

and independently for the groups of patients with methylated and unmethylated MGMT. 219

In addition, p-values derived from Mann-Whitney U test or Fisher exact test are included.’ And simply write Table 1 in brackets after the first sentence (line 217).  Similar to what you did in 3.2.2 – this is much better.

Line 225: the p value statement is incorrect – the sign is the wrong way round; should be p<0.05.

Discussion:

Line 333: ‘metanalysis’ – should be ‘meta-analysis‘

Line 337: remove ‘can’ after ‘may’

Line 338: remove ‘get’ before ‘benefit’

Line 350: add in ‘a’ before ‘different number’

Line 358: ‘prolonged’ should be ‘prolong’

Reviewer 3 Report

Unfortunatelly, this Research does not bring anything novel in the field of glioblastoma, therefore 

Reviewer 4 Report

Paper titled (Lack of benefit of temozolomide for MGMT methylated patients with high vascular glioblastomas: a confirmatory study.) by Torrez et al. discussed the benefit of using temozolomide in 4 different scenarios to judge the max efficacy. This is an interesting paper with high clinical potential however, I think some revisions are necessary::

1- Summary, clarify what is HGMT at first appearance And also in abstract

2- Introduce HGMT in the introduction to a deeper extent a

3- How patients were identified HGMT methylated or not methylated? this is not clear in methods or other parts

4-Figure 3: symbols are not clear & need to be identified in the figure legends. What  * means or what  ** means? so the figure becomes self explanatory. 

5- SOme statements are overstated, please revise the conslusions to be more logic.

Reviewer 5 Report

Concise summary

 The authors aimed to demonstrate the lack of benefit of temozolomide for patients MGMT methylated with high vascular glioblastomas (GBMs). Methodologically, it is a combination of quantification of the GBMs vascularity based on dynamic radiological parameters and evaluation of MGMT status methylation. This vascularity is understood in terms of blood volume, flow blood and edema. The cut-points to divide GBMs is obtained considering the median of several references. It is studied the correlation between MGMT status methylation, number of cycles of temozolomide and tumor vascularity. The authors conclude that MGMT methylation status and moderate vascularity of the GBMs correlates with better prognosis, but not with high vascularity. In addition, the authors show that the effect of temozolomide is better in MGMT methylated GBMs with moderate vascularity. The conclusions are enough clear and are supported by the results. Furthermore, it is postulated that the analysis of these biomarkers would help to identify patients who are most likely to benefit from a higher number of temozolomide cycles.

.

Final considerations

 Major criticisms

 1.Title:

  1.1 The sentence “…. MGMT methylated patients …” is not correct. The patients are not methylated, but only their tumors. This sentence should be conveniently modified.

 1.2. The title is not informative enough. As it is seen in “Conclusions” the objectives of the work have been double: the evaluation of the MGMT methylation status and the number of cycles of temozolomide according to the radiological definition of tumor vascularity. However, the conclusion about temozolomide effect and vascularity is not shown. By the way, it would be more informative to entitle the article including the  correlation between MGMT status and moderate vascularity.

 2.Methods (lines 96-97):

 2.1. The methodology used to test the MGMT status is not sufficiently explained. It would be important to know which was the applied methodology. Which was the selected MGMT methylation cut-off to classify the GBMs as methylated vs non-methylated?

 2.2. As it is a multicentric study, it would be relevant to know which was the applied methodology in the molecular laboratories?

 2.3 The intensity of tumor vascularization is obtained through MRI-based validated quantification procedure, In addition, it is shown that the tumor vascularity is heterogeneous with high and low angiogenic areas (figure 1). How the authors define the final vascularization in each case?

 3.Ressults/Discussion.

 3.1. A correlation between MGMG status and tumor vascularity is shown. It is only a descriptive finding. Could the authors hypothesize a biological rationale between both findings?

 3.2. It is known the correlation between MGMT status and temozolomide.

Finally, the article addresses a relevant clinical issue through an interesting clinical study. The results are relevant in terms of data to predict that patients with GBM could be treated greater quantity of temozolomide. The results are relevant and could have a clinical impact.

Round 2

Reviewer 4 Report

1- The new title is very long and not the best one, please revise:

Lack of benefit of extending temozolomide in patients with methylated MGMT and high vascular glioblastomas: a confirmatory study of the combined effect of MGMT methylation and tumor vascularity: is "methylated" is necessary? 

2- The second part of the title is useful? does it add something?

please reformulate it as written in the study aim.

3-Authors wrote in figure legends:*Data points beyond the whiskers are 351
shown with +.: this does not make sense: should be *: significantly different from ...group.

4-Figure resolution is still poor & symbols are very small

5-Incresase thickness of axis of all figures & darken the color

6- Conclusion: In conclusion, our results demonstrate the lack of benefit of presenting methylated MGMT : please revise

7- ensure expression of terms is the same allover the manuscript & the conclusion & aim are the same way in all paper sections.

8- write a separate section at the end of the methods on "statistical analysis" in which, wrirte everything about statistics & curves

Reviewer 5 Report

The authors aimed to demonstrate the lack of benefit of temozolomide for patients MGMT methylated with high vascular glioblastomas (GBMs).  This vascularity is understood in terms of blood volume, flow blood and edema. The authors have corrected the text according to the reviewer's suggestions. Although it is a descriptive article based on radiological findings without a biological rationale supporting the conclusions, the work has enough originality and quality.
